# Association of Gut Microbiome with Muscle Mass, Muscle Strength, and Muscle Performance in Older Adults: A Systematic Review

**DOI:** 10.3390/ijerph21091246

**Published:** 2024-09-20

**Authors:** Martin Hubert Mayer, Selam Woldemariam, Christoph Gisinger, Thomas Ernst Dorner

**Affiliations:** 1Karl Landsteiner University of Health Sciences, 3500 Krems an der Donau, Austria; 2Karl Landsteiner Institute for Health Promotion Research, 3062 Kirchstetten, Austria; 3Center for Public Health, Department for Social and Preventive Medicine, Medical University of Vienna, 1090 Vienna, Austria; 4Academy for Ageing Research, Haus der Barmherzigkeit, 1160 Vienna, Austria

**Keywords:** muscle mass, muscle strength, muscle performance, microbiota alterations, sarcopenia

## Abstract

Sarcopenia, characterized by reduced muscle mass, strength, or performance, is a common condition in older adults. The association between the gut microbiome and sarcopenia remains poorly understood. This systematic review aims to evaluate the relationship between muscle parameters and the intestinal microbiome. A systematic search was conducted in PubMed, EMBASE, Cochrane Library, and Google Scholar for studies published between 2002 and 2022 involving participants aged 50+. Studies were included if they assessed sarcopenia using at least one measure of muscle mass (skeletal muscle mass, bioelectrical impedance analysis, MRI), muscle strength, or muscle performance (SARC-F questionnaire, Timed-Up-and-Go Test, Chair Stand Test, grip strength, gait speed, Short Physical Performance Battery, 400 m Walk Test). The microbiome was measured using at least RNA/DNA sequencing or shotgun metagenomic sequencing. Twelve studies were analyzed. Findings revealed that a higher abundance of bacterial species such as *Desulfovibrio piger,* and *Clostridium symbiosum* and reduced diversity of butyrate-producing bacteria was associated with sarcopenia severity, as indicated by decreased grip strength, muscle mass, or physical performance. The gut microbiome plays a significant role in age-related muscle loss. Probiotics, prebiotics, and bacterial products could be potential interventions to improve muscle health in older adults.

## 1. Introduction

European countries have seen a decline in mortality rates and increased life expectancy, which has led to an aging society. According to Eurostat, the EU population aged 65 and older is projected to almost double from 17% in 2010 to 30% by 2060 [1]. This demographic shift presents a challenge for individuals and society as a whole; therefore, it is imperative to understand the complex health conditions associated with aging. Sarcopenia, characterized by age-related decline in muscle mass and function, significantly increases the risk of adverse outcomes such as impaired quality of life, falls, hip fractures, frailty, hospitalization, disability, and mortality [2,3,4,5,6,7]. Individuals with sarcopenia have a higher likelihood of premature mortality compared to their age-matched counterparts [8]. Sarcopenia also strongly predicts disability, defined as limitations in activities of daily living (ADL) and care home admissions [2]. However, it is plausible that age-associated modifications in the gut microbiota and muscle tissue composition could be driven by shared underlying processes, such as chronic inflammation, dysfunction of the immune system, and changes in hormone levels, which could influence both microbiota alterations and the onset of sarcopenia. Beyond health-related implications, sarcopenia imposes social, medical, and economic burdens. In the UK, the annual cost for a hospitalized sarcopenia patient was estimated at GBP 2707 in 2018 [9]. Given its broad impact and disabling complications, implementing preventive interventions is crucial to promote healthy aging and reverse functional declines in older individuals.

Currently, there is no single effective therapeutic intervention for sarcopenia. This multifactorial condition can arise from immunological, hormonal, neurological, inflammatory, nutritional, physical activity (PA), and lifestyle changes [4,10]. A study estimates that aging atrophy reduces muscular strength, with a likely decrease of 12% to 14% per decade after age 50, due to a reduction in muscle fibers and atrophy of type II muscle fibers [11]. However, several intervention studies show that resistance exercise can improve muscle mass and function in individuals aged 60 and older [5,12,13]. Additionally, others have explored the impact of nutrition on muscle parameters [14,15]. A recent review [15] found that daily dietary supplements, including essential amino acids, antioxidants, polyunsaturated fatty acids, minerals, vitamin D, and physical exercise, benefit muscle mass and function in healthy individuals aged 60 and older.

The ‘gut–muscle axis’, which explores how the gut microbiome affects muscle mass, strength, and function in older adults, plays a crucial role in both the prevention and management of sarcopenia [16]. Previous research indicates that diet composition and the gut microbiome change with age and are correlated with muscle mass decline, thereby impacting physical performance [17,18]. Advanced age leads to gut microbiome dysbiosis, characterized by altered microbial diversity, predominant bacteria, and reduced beneficial bacterial metabolites [19]. These biological processes, particularly those related to inflammation and the immune system, are greatly influenced by the gut microbiome. Studies show differences in gut microbiome composition, measured using RNA (rRNA) sequencing, between older and younger participants, marked by variations in alpha diversity (species diversity within a sample) and beta diversity (species diversity between samples) [20,21,22,23]. Animal studies support these findings, suggesting that variations in the gut microbiome and metagenome influence biological processes such as inflammation, nutrient bioavailability, and lipid metabolism, contributing to age-related muscle decline [24,25]. Further understanding of the complex molecular mechanisms of this ‘gut–muscle axis’ is needed. Although human data are limited, animal studies provide significant insight into the gut microbiome’s development, composition, stability, and role in various pathological processes.

In light of the aging population in Europe, this review provides insight into potential novel interventions for sarcopenia prevention and treatment. Hence, this systematic review aims to identify the gut–muscle axis mechanisms involved in age-associated sarcopenia by evaluating clinically relevant parameters of the association between sarcopenia—measured by muscle mass, muscle strength, or muscle performance—and the gut microbiome. Specifically, to identify whether individuals with sarcopenia show alterations in their microbiome or if changes in the microbiome can lead to sarcopenia. The review addresses the question: What are the associations between muscle mass, muscle strength, or muscle performance and the gut microbiome among individuals aged 50 years or older? These associations may involve muscle parameters affecting the microbiome, the microbiome affecting muscle parameters, or both being influenced by a third factor, such as nutrition.

## 2. Materials and Methods

### 2.1. Search Strategy 

A systematic search was performed by M.H.M. using four databases: PubMed, EMBASE, the Cochrane Library, and Google Scholar. The search focused on peer-reviewed primary studies written in English or German and published from 2002 onward to ensure up-to-date information. The search strategy combined free-text terms and medical subject headings, adapted to each database using Boolean operators AND and OR. The search strings on PubMed included the following terms: (“sarcopenia”[MeSH Terms] OR “sarcopenia”[All Fields] OR “SARC-F”[All Fields]) OR (“muscles”[MeSH Terms] OR “muscles”[All Fields] OR “skeletal muscle mass”[MeSH Terms] OR “SMM”[All Fields] OR “muscle function”[MeSH Terms] OR “muscle quantity”[All Fields] OR “muscle quality”[All Fields] OR “muscle strength”[MeSH Terms] OR “physical performance”[All Fields]) AND (microbiome[MeSH Terms] OR microbiome[All Fields] OR microbio*[All Fields]). Further details about the search strings and terms used, as well as the search strings for databases other than PUBMED, can be found in the supporting materials (eSearch String). This study was prospectively registered in December 2022 with the International Prospective Register of Systematic Reviews (PROSPERO) under the unique identification code CRD42022379280. Papers were searched in databases between July and October 2022.

### 2.2. Exclusion and Inclusion Criteria

Articles were eligible if they met at least one of the following criteria: (a) human studies focused on participants aged 50 years or older; (b) measured muscle strength and performance using at least one measure of the SARC-F (Strength, Assistance in walking, Rise from a chair, Climb stairs, and Falls) questionnaire, Timed-Up-and-Go Test (TUG), Chair Stand Test (CST), grip strength, gait speed, Short Physical Performance Battery (SPPB), or 400 m Walk Test [26]; (c) measured muscle mass/quantity using at least skeletal muscle mass (SMM), computed tomography (CT), ultrasound (US), bioelectrical impedance analysis (BIA), magnetic resonance imaging (MRI), or dual-energy X-ray absorptiometry (DXA); (d) measured microbiome using either RNA- or DNA-sequencing or shotgun metagenomic sequencing; (e) cross-sectional studies, case–control studies, cohort studies, randomized and non-randomized controlled trials; and (f) written in English or German.

The exclusion criteria were: (a) animal studies or in vitro models; (b) reviews, editorials, commentaries, and other studies without primary data; (c) conference abstracts, study protocols, or congress presentations without clear information about the outcome, design, or methods; and (d) studies written in languages other than English or German.

### 2.3. Study Selection

Following the Preferred Reporting Items for Systematic Reviews and Meta-Analyses (PRISMA) guidelines [27], all identified references were imported into EndNote^®^ to be screened for duplicates and suitability by M.H.M. and S.W. The screening process was conducted independently by two authors (S.W. and M.H.M.) to reduce bias. In case of disagreements, a third author (T.E.D.) made the final decision on inclusion. First, the titles and abstracts of the identified references were assessed for duplicates and relevance to the topic. Second, full-text articles were evaluated based on the inclusion and exclusion criteria.

### 2.4. Quality Assessment

The quality of randomized controlled trials (RCTs) was assessed using the Cochrane Risk of Bias 2 (RoB) tool, which evaluates bias across five domains: (1) randomization process, (2) deviations from intended interventions, (3) missing outcome data, (4) outcome measurement, and (5) selection of the reported result [28]. Each domain includes three to seven signaling questions, answered as “Yes”, “Probably yes”, “Probably no”, “No”, or “No information”. These answers are used to derive a risk of bias judgment for each domain: “low risk of bias”, “some concerns”, or “high risk of bias”. An overall risk of bias judgment is then made based on the domain judgments.

The Newcastle–Ottawa Scale (NOS) was used to assess the methodological quality of non-randomized studies by M.M. and S.W. The NOS evaluates bias in three domains: (a) selection of participants, with four items assessing representativeness, selection of the non-exposed cohort, ascertainment of exposure, and demonstration that the outcome was not present at the study’s start; (b) comparability, with one item assessing design or analysis; and (c) outcome, with three items assessing outcome assessment, follow-up duration, and adequacy of follow-up. A maximum of one star can be awarded for each item in selection and outcome, and two stars for comparability. The highest score is 9, indicating high quality. Scores of 0–3, 4–6, and 7–9 indicate low, moderate, or high quality, respectively. A median score of ≥4 points was considered high quality [29].

## 3. Results

A total of 800 studies were identified through the electronic database search. After removing duplicates, 658 studies remained for title and abstract screening. Of these, 599 records were excluded, leaving 59 studies for full-text screening. From these, 13 studies involved participants aged 50 years or younger, 2 studies were conducted on animals, 7 studies were unpublished clinical trials, 19 were excluded due to study type without peer-reviewed publications, 3 studies did not use RNA- or DNA-sequencing to measure the gut microbiome, 1 study did not clearly state the outcome measure we were looking for, and 1 study lacked clear measurement criteria for muscle mass, strength, or performance. Finally, 12 studies were included in the synthesis for methodological quality according to the NOS and RoB2. All included studies in the review were considered qualitatively suitable. Figure 1 shows the PRISMA flow chart of study selection.

### 3.1. Charcterstics of Included Studies

The study characteristics of the included studies are described in Table 1. All studies were published within the last four years, from 2019 to 2022. Four studies were conducted in Italy, three in China, two in Japan, and one each in South Korea, Taiwan, and the UK. Cohort sample sizes ranged from 6 [30] to 1417 individuals [31]. Most studies focused on individuals older than 60 years [23,30,32,33,34,35,36,37,38] with two including participants from 50 years of age [22,31]. Five studies included community-dwelling individuals, one focused on hospitalized patients before cardiac surgery, and one included hospitalized and geriatric outpatients. Six studies used BIA to measure muscle mass, five used DXA, and one used SMI. Handgrip strength was the most common measure of muscle strength [23,30,31,33,34,39], followed by SPPB and the Chair Stand Test. Other measurements included gait speed, the Sit-And-Reach Test, five-time Chair Stand Test, 6 m Walk Test, and 400 m Walk Test. All studies used established homogenous cut-off values for low physical strength, function, and mass, such as the definition of the European Working Group on Sarcopenia in Older People 2. One study [22] did not specify muscle strength/function measurements. The most common method for assessing the gut microbiome was 16S rRNA gene sequencing, with only two studies using shotgun metagenomic sequencing [31,37].

### 3.2. Key Findings

Most studies found an association between sarcopenia—measured by muscle mass, strength, and performance—and alterations in the gut and fecal microbiome. One study [30] indicated that prebiotics, which increase gut microbiome abundance, positively affected skeletal muscle mass.

Han and colleagues [33] found significant correlations between specific gut bacteria (such as a decrease in *Marvinbryantia*) and fecal butyrate levels in participants with low muscle mass (LM), both positively associated with low muscle mass. Kang and colleagues [34] observed significantly reduced bacterial profiles and diversity in "Case" and "pre-Case" groups with reduced muscle mass and function. At the genus level, for example, *Lachnospira* was significantly reduced, while *Lactobacillus* was more abundant in these groups. Ticinesi and colleagues [37] reported that subjects with reduced lower limb function and muscle mass had significantly lower amounts of certain bacteria, such as *Roseburia inulinivorans*, while the amount of another bacteria, *F. prausnitzii*, was five times higher in those with normal lower limb function and muscle mass. Wang and colleagues [31] found that sarcopenic participants had higher abundances of seven bacteria, such as *Desulfovibrio piger,* all associated with sarcopenia severity.

Furthermore, two studies [33,38] noted decreased alpha diversity in sarcopenic patients. At the phylum level, sarcopenic patients had reduced *Proteobacteria* and enriched *Firmicutes* and *Bacteroidetes.* At the genus level, they exhibited a higher *Prevotella/Bacteroides* ratio and more *Coprococcus*, with varying amounts of *Lachnospiraceae.* A high *Prevotella/Bacteroides* ratio and specific levels of *Coprococcus* and *Lachnospiraceae* could distinguish sarcopenic participants from controls. Wu and colleagues [38] concluded that a high *Prevotella/Bacteroides* ratio (over 1.7) and the relative amounts of *Coprococcus* (1.00–3.70%) and *Lachnospiraceae* (0.00–1.68%) could be used to distinguish participants with sarcopenia. By contrast, three studies [23,31,36] reported no significant difference in alpha diversity between older adults with sarcopenia (PF&S) and non-sarcopenia groups.

### 3.3. Findings in the Context of Disease

Similar results were found in three studies with chronically ill patients [22,35,39]. One study [35] found greater abundances of the families *Micrococcaceae* and *Verrucomicrobiaceae* and lower abundances of the families *Veillonellaceae* and *Gemellaceae* in sarcopenic patients with chronic kidney disease. In another study with liver cirrhosis patients [39], significantly reduced alpha diversity was observed in sarcopenic patients compared to their non-sarcopenic peers. Sarcopenic patients also had lower abundances of *Prevotella, Methanobrevibacter,* and *Akkermansia,* and a higher relative abundance of *Eggerthella.* Yamamoto et al. [22] found that, in patients with chronic liver disease and low muscle mass, alpha diversity was significantly lower. At the phylum level, *Proteobacteria* and *Bacteroidetes* were higher and *Firmicutes* was lower in the low SMI group. At the genus level, *Coprobacillus, Catenibacterium,* and *Clostridium* were lower, while *Bacteroides* was higher. The *Firmicutes* to *Bacteroidetes* ratio was significantly higher in the low SMI group.

### 3.4. Findings in the Context of Nutrition

Two studies [30,32] examined the associations between nutrition, gut microbiome, muscle mass, and strength. Cox [32] and Tominaga [30] found that nutrition played an important role in bacterial diversity in older adults. Cox and colleagues [32] reported that poor appetite was significantly associated with changes in microbiome diversity and abundance, as well as reduced muscle mass and strength, indicated by higher mean chair stand time. Tominaga and colleagues [30] investigated whether a prebiotic (*1-kestose*) could alter the microbiota composition and aid in muscle recovery in elderly patients with sarcopenia. They found an increase in *Bifidobacterium*, particularly *Bifidobacterium longum*, but no significant change in alpha diversity. Significant improvements in SMI, SMM, TM, and TMM were observed after 12 weeks, suggesting that *1-kestose* can positively affect microbiome composition and aid in muscle recovery in sarcopenic patients. 

## 4. Discussion

In this systematic review, 12 human studies were included to investigate the association between muscle mass, strength, and/or performance and the gut microbiome. Ten observational studies [22,23,31,33,34,35,36,37,38,39] investigated the differences in gut microbiome composition between participants with and without changes in muscle mass/strength and/or performance. One study examined the effect of prebiotic interventions on frail patients aged 80 and older [30], and one study explored the impact of diet on gut microbiome diversity and muscle strength among women [32]. Overall, six studies [22,23,30,31,34,35] found significant changes in the gut microbiome composition in patients with low muscle mass and/or strength compared to healthy controls, while four studies reported no significant changes [36,37,38,39].

Interestingly, similar findings were reported in three observational studies involving patients with chronic kidney disease [35], liver cirrhosis [39], and chronic liver disease [22]. In the presence of sarcopenia in older patients, variations in the gut microbiome suggest its involvement in regulating muscle physiology in various muscle-wasting diseases. Although no differences in alpha diversity (species diversity within a sample) were reported in chronically ill patients, reduced abundances of butyrate-producing bacteria such as *Prevotella*, *Faecalibacterium prausnitzii*, and *Roseburia* were observed, which are responsible for maintaining healthy muscle status through SCFA production [35]. However, in chronically ill patients, the observed alteration of the gut microbiome may be influenced by the underlying disease and associated medication use. On the other hand, other studies [22,31,35,36] have reported higher abundances of bacteria such as *Desulfovibrio piger* and *Clostridium symbiosum* in sarcopenic participants, which have been linked to increased sarcopenia severity. The heterogeneity in these findings indicate that further research is needed to confirm the specific changes in the gut microbiota composition that may modulate inflammation and muscle mass, muscle strength, and muscle performance in aging.

The pathogenesis of sarcopenia involves inflammation, increased oxidative stress, poor mitochondrial function, and anabolic resistance [10,40,41,42,43,44]. It is not only associated with aging [42] but also with chronic conditions such as type II diabetes and chronic kidney disease [43,44]. Considering the individual differences in the gut microbiome and the possible confounding factors of chronic diseases, further studies with larger sample sizes are needed to increase the robustness of these findings. Given the cross-sectional design of the included studies, their findings are descriptive. Furthermore, this review includes intervention studies that evaluate multiple outcomes, such as muscle mass, muscle strength, and muscle performance—all of which are characteristics of sarcopenia. These outcomes are highly interrelated, but each measures a different aspect of sarcopenia. The findings suggest that differences in gut microbiota signatures may be influenced by the specific clinical outcomes being studied. Therefore, longitudinal cohort studies are needed to clarify the potential causal relationship between the gut microbiome and changes in individual clinical parameters of sarcopenia in older adults. Finally, more research into the mechanisms of gut dysbiosis and age-related muscle wasting is crucial. Identifying bacterial phyla associated with advanced sarcopenia could potentially serve as indicators for future clinical sarcopenia diagnosis.

Various interventions have targeted the gut microbiome to strengthen muscle mass and function, but the role of the diet in these changes remains unclear. A UK study [32] examined the effect of diet on the gut microbiome in 102 community-dwelling older adults aged 65 and older. The study found a reduction in bacterial richness and diversity, particularly *Lachnospira*, a butyrate-producing bacteria, among those with poor appetite. However, a similar study in the Netherlands [45] found no difference in alpha or beta diversity, or bacterial abundance between diet and control groups. These differences may stem from differences in participant stratification, which was based on isocaloric dietary interventions rather than baseline microbiome profiles. Furthermore, it is worth noting that increasing one macronutrient can result in a relative decrease in others, thus influencing outcomes.

Recent studies have investigated the effects of diet, protein supplements, and exercise on muscle mass and the gut microbiome to better understand the relationship between metabolites and disease [30,46,47,48,49,50,51,52,53]. An observational study [30] found that *1-kestose* supplementation in older sarcopenic patients increased *Bifidobacterium longum* in the gut and improved muscle recovery. A randomized trial [46] found that a high-protein diet, with or without prebiotic supplements, improved lean muscle mass in older women 65 years and older but did not significantly change gut bacterial diversity. However, the 18-week study period may have been too short, and exercise could have influenced the results. Nevertheless, similar findings on the association between prebiotic supplements and bacterial diversity in the gut have been reported in other studies [49,50,51,52], but the causal link between inulin supplements and muscle growth remains unclear. Other studies [45,46,53] explored dietary protein interventions, comparing plant-source proteins and daily meat intake, and found that the microbiome’s response to plant-based proteins may differ, potentially leading to qualitative malnutrition, with reductions in specific macronutrients or micronutrients affecting muscle. Exercise is also reported to positively affect the gut microbiome and can offer new interventions to treat sarcopenia [50,51,52]. Therefore, for effective, personalized dietary and lifestyle interventions, future studies should examine these distinctions and monitor long-term microbiome changes to better understand protein intake’s impact on physical functioning.

This is the first systematic review focusing on the association between the gut microbiome and muscle, particularly in sarcopenic patients. The study includes clinical and non-clinical studies from available databases using a reproducible search approach. Our study summarizes and analyzes the current literature, highlighting key findings and issues in this field. We also included patients with chronic conditions to explore potential mechanisms with current evidence. By employing various measurements of sarcopenia, we aimed to capture the broadest range of studies, and our inclusion of recent studies from 2002 onward is another strength of the review. However, there are several limitations to this review. The limited number of eligible studies resulted in heterogeneity of findings, making it difficult to draw definitive conclusions. Moreover, we excluded animal studies, which might have provided additional insights into the topic. The small sample sizes in clinical studies also pose a limitation that future research should address. Furthermore, only two of the included studies examined nutrition and diet as potential confounding factors in the association between sarcopenia and the gut microbiome in older adults. This highlights a gap in the literature regarding the comprehensive assessment of potential confounders such as physical activity, medication use, or other relevant lifestyle factors. Future research should account for potential confounders to better understand the complex relationship between clinically relevant parameters of sarcopenia—measured by muscle mass, muscle strength, or muscle performance—and the gut microbiome in older adults. Another possible limitation of the systematic review is that the included original papers were limited to a few publication languages (English and German).

It should be noted that the present study differs from a previous study that examined the relationship between the gut microbiota and sarcopenia, which included young adults, animal studies, and fecal transplant studies [54]. The previous study aimed to determine if gut microbiota directly affects muscle mass and function by identifying beneficial bacteria and bacterial products and understanding how the gut microbiome regulates muscle. By contrast, the present study focused primarily on human participants older than 50 years to address geriatric sarcopenia and provide clear clinical insights. Additionally, our study examined the association between the gut microbiome and muscle parameters in the context of chronic conditions like chronic kidney disease, chronic liver disease, and liver cirrhosis patients, which were not considered in the previous review. Our study design and findings offer novel insight into the alterations of the gut microbiome in the aging population.

## 5. Conclusions

The present study indicates that the association between clinically relevant parameters of sarcopenia—measured by muscle mass, muscle strength, or muscle performance— and the gut microbiome in older adults aged 50 and older remains unclear with no consistent pattern. Some studies have reported reductions in gut microbiome diversity and specific bacterial species in those with low muscle mass, strength, and/or performance, while others have found higher abundances of certain bacterial species in older adults. As Europe’s population ages, understanding the genesis of sarcopenia is crucial for developing targeted therapeutic interventions to delay age-related muscle dysfunction.

## Figures and Tables

**Figure 1 ijerph-21-01246-f001:**
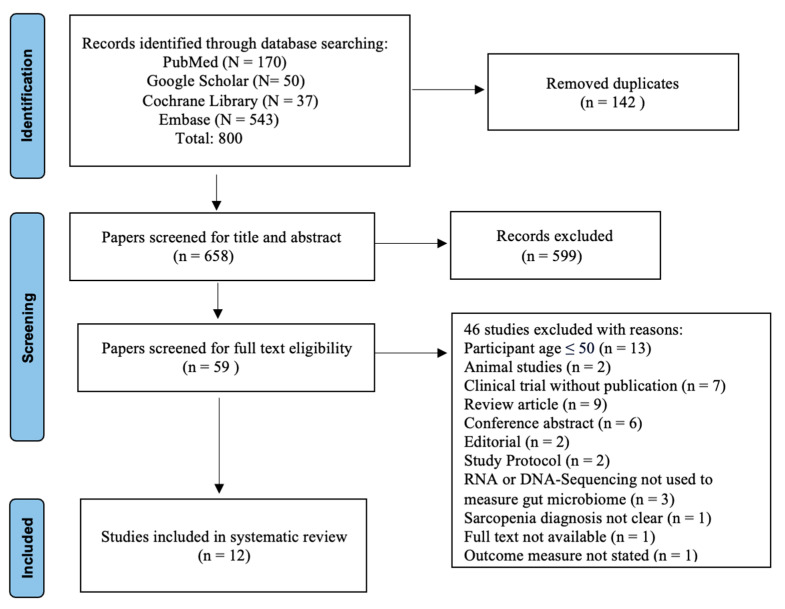
PRISMA flow diagram.

**Table 1 ijerph-21-01246-t001:** Characteristics of included studies.

Author [Ref]	Study Participants	Sample Size	Study Design	Muscle Mass/Strength Measurement	Gut Microbiome Measurement	Association between Muscle Mass/Strength and Gut Microbiome	Key Findings	NOS
Yamamoto et al., 2022 [22]	≥50 years, 63% female, Japan, CLD patients	69	Cross-sectional	SMI/ Not assessed	16S rRNA Gene Sequencing	In the low SMI group, *Proteobacteria* and *Bacteroides* were significantly higher, while phylum *Firmicutes* was lower. At the genus level, *Coprobacillus*, *Catenibacterium*, and *Clostridium* were lower, and Bacteroides were higher. The *Firmicutes* to *Bacteroidetes* ratio was higher in the low SMI group	Alpha diversity and specific bacterial species were significantly lower in low SMI group	9 stars
Lee et al., 2022 [23]	≥60 years, 75% female, South Korea	60	Cross-sectional	BIA/grip strength, 6 m Walk Test	16S rRNA Gene Sequencing	*Anaerotruncus* and *Phascolarctobacterium* sp. are significantly correlated with sarcopenia at the genus level. *Prevotella* abundance was significantly negatively associated with sarcopenia	Diversity (at the species level) differed between case and control groups	8 stars
Tominaga et al., 2021 [30]	≥82 years, 83% female, Japan	6	Cohort	BIA, SMI, SMM, TMM/hand grip strength, gait speed	16S rRNA Gene Sequencing	A prebiotic administration of 1-*kestose* increased the Skeleton Muscle Index (SMI) significantly after 12 weeks	Prebiotic administration increased the population of *B. longum* and muscle mass	8 stars
Wang et al., 2022 [31]	≥50 years, 59% female, China	1417	Cross-sectional	BIA/grip strength, SPPB, Chair Stand Test, gait speed	Shotgun Metagenomic Sequencing	Sarcopenia participants showed significant higher abundances of *Desulfovibrio piger*, *Clostridium symbiosum*, *Hungatella effluvii*, *Bacteroides fluxus*, *Absiella innocuum*, *Clostridium citroniae*, and *Coprobacter secundus*. These species were also associated with sarcopenia severity	Specific microbiome alterations in sarcopenia participants. Change in beta diversity	8 stars
Cox et al., 2021 [32]	≥65 years, 95% female, UK	204	Cross-sectional	DXA/CST	16S rRNA Gene Sequencing	Participants with poor appetite showed significant differences in bacterial taxa, alpha and beta diversity, and reduced muscle strength	Poor appetite was significantly associated with bacterial diversity and muscle mass	8 stars
Han et al., 2022 [33]	≥65 years, 68% female, Taiwan	88	Cross-sectional	BIA/grip strength, gait speed, Sit-And-Reach Test	16S rRNA Gene Sequencing	*Marvinbryantia*, *Akkermansia*, *Ruminococcaceae UCG-10*, *Subdoligranulum*, *Barnesiella* sp., and *F. prausnitzii* were significantly reduced in low muscle group. Fecal butyrate was significantly correlated with skeletal muscle mass	Specific members of gut microbiota are significantly correlated with low muscle mass	8 stars
Kang et al., 2021 [34]	≥60 years, 58% female, China	87	Cross-sectional	BIA/grip strength, Chair Stand Test	16S rRNA Gene Sequencing	Participants with low muscle mass showed significantly reduced species diversity in Chao1 index	Differences in microbiota between case and control groups	7 stars
Margiotta et al., 2021 [35]	≥65 years, 69% male, Italy, CKDpatients	64	Cross-sectional	Not Specified/Not specified	16S rRNA Gene Sequencing	Cases showed high abundances of *Micrococcaceae*, *Verrucomicrobiaceae*, *genera Megasphaera*, *Veillonella*, *Rothia*, *Coprobacillus*, and *Akkermansia*. Lower abundances of *Veillonellaceae* and *Gemellaceae*	The gut microbiota composition is significantly different between sarcopenic and not-sarcopenic individuals with CKD	7 stars
Picca et al., 2019 [36]	≥70 years, 57% male, Italy	35	Cross-sectional	DXA/SPPB, 400 m Walk Test	16S rRNA Gene Sequencing	Sarcopenic cases showed significant increases in *Peptostreptococcaceae*, *Bifidobacteriaceae*, *Dialister*, *Pyramidobacter*, and *Eggerthella*, and significant depletions in *Slackia* and *Eubacterium*. No significant difference in alpha diversity was found	Four microbes (*Barnesiellaceae*, *Christensenellaceae*, *Oscillospira*, *Ruminococcus*) played a role in participants with sarcopenia	8 stars
Ticinesi et al., 2020 [37]	≥70 years, 82% female, Italy	17	Cross-sectional	BIA/SPPB, grip strength, gait speed, Chair Stand Test	Shotgun Metagenomic Sequencing	Sarcopenic cases had significantly lower amounts of *Roseburia inulinivorans* and *Alistipes shahii. F. prausnitzii* was five times higher in non-sarcopenic subjects	Sarcopenic subjects displayed different fecal microbiota compositions at the species level	8 stars
Wu et al., 2022 [38]	≥65 years, 54% female, China	192	Cross-sectional	Not assessed/SPPB	16S rRNA Gene Sequencing	A high *Prevotella/Bacteroides* ratio (over 1.7) and relative amounts of *Coprococcus* (1.00–3.70%) and *Lachnospiraceae* (0.00–1.68%) were observed in sarcopenic participants	Both the abundance and diversity of the gut microbiota were low in sarcopenic participants	8 stars
Ponziani et al., 2021 [39]	≥58 years, 66% male, Italy, liver cirrhosis patients	100	Cross-sectional	DXA/grip strength	16S rRNA Gene Sequencing	Significantly low abundances of *Prevotella*, *Methanobrevibacter*, and *Akkermansia* and a significant higher relative abundance of *Eggerthella* were found in participants with sarcopenia and liver cirrhosis	Participants with sarcopenia and liver cirrhosis showed a significant reduced alpha diversity and gut microbiota composition	9 stars

Note: BIA = bio impedance analysis; CKD = chronic kidney disease; CLD = chronic liver disease, CST = Chair Stand Test; DXA = dual-energy X-ray absorptiometry; SPPB = Short Physical Performance Battery; SMI = skeletal muscle index; SMM = skeletal muscle mass; TMM = total muscle mass; NOS = Newcastle Ottawa Scale; Significance indicated by *p* < 0.05.

## Data Availability

Further details on data and analytical methods can be obtained from the corresponding author upon request. The protocol for the conducted research was pre-registered with PROSPERO (https://www.crd.york.ac.uk/prospero/display_record.php?RecordID=379280, accessed on 19 September 2024).

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
