# Peer review of "Association of Gut Microbiome with Muscle Mass, Muscle Strength, and Muscle Performance in Older Adults: A Systematic Review"

_ijerph, 2024, doi:10.3390/ijerph21091246_

Round 1

Reviewer 1 Report

Comments and Suggestions for Authors

This study is a systematic review on the 'Association of Gut Microbiome with Muscle Mass, Muscle Strength, and Muscle Performance in Older Adults.'

The study has three major flaws. The first relates to its innovation. Several reviews published in the last five years have addressed this topic or similar ones (for example: DOIs: 10.3390/nu11071633 / 10.3390/nu13062045). The introduction fails to contrast with the already published reviews and demonstrate the primary innovation that this study aims to achieve.

The second flaw is the absence of a meta-analysis. Given the title, this is what is expected.

While the first two points can be addressed through revisions, the third and most significant cannot. The screening process in this systematic review is flawed. Initially, articles in German were considered, but why were articles in Italian, Spanish, Japanese, and other languages not included? This is a serious issue that should not have occurred. The authors will likely respond that this was done to increase the number of selected articles, but this would be a rhetorical response if that is their answer to my question. Furthermore, the authors state 'published from 2002 onwards to ensure up-to-date information.' This makes no sense for two related reasons. If the goal is to provide new information, 2002 may already be outdated. Secondly, if articles in languages other than English were included to increase the number of retrieved articles, why impose a publication year limit? Additionally, the exact strategy for all databases used was not provided, only for PUBMED. The term 'for example' was also used. The authors certainly did not use the same structure for the other databases."

Reviewer 2 Report

Comments and Suggestions for Authors

This is a thorough analysis of the published reports of decline in muscle function and muscle mass in people over the age of 50 and in the same subjects studied, changes in the species profile of the gut microbiome. The implication in this review of the literature is that there is some causal link between the microbiome changes and the development of sarcopenia. It is understandable that speculation on what that causal link might be is unwarranted in the absence of indicative findings. However, the alternative viewpoint needs to be discussed, that the muscle changes and the microbiome changes are parallel modifications related to the process of aging without a causal link between them. If there were a causal link, it would presumably be from some metabolites of particular microbial species that are absorbed across the large intestinal mucosa and then have some modifying effect on the immunological, hormonal, neurological and inflammatory factors (lines 40-41) that are associated with the onset of sarcopenia. It would therefore be helpful if such suggestions were to be specified in the text in order to interpret the general principles of the positive associations summarised in this review.

A second modification that would be of considerable help in understanding what this review has identified, would be to have a summary table of the key findings reported at lines 180 to 206. It is difficult to grasp the findings with the many microbial species simply listed in the text, with the strength of the evidence for these changes linked to the evidence of sarcopenia.

Minor changes:

Line 163: “…and one in South Korea, Taiwan, the UK and the USA.” This would be more understandable if the word “each” were inserted: “…and one each in South Korea, Taiwan ….etc.”

Table 1. It would be helpful if the abbreviations in the column heading Nos/Ros were to be  explained in the legend to this Table. Presumably, Nos refers to Newcastle Ottawa Scale. If so, shouldn’t Ros be RoB – risk of bias?

Comments on the Quality of English Language

The use of English is well done

Reviewer 3 Report

Comments and Suggestions for Authors

The present study presents a systematic review of studies investigating the relationship between the gut microbiome and markers of muscle mass, physical function and performance. Generally, the manuscript is well-written and a well-conducted review but below is a summary of suggestions for the author’s to consider.

Abstract

The abstract is missing important key findings such as that lower levels of butyrate producing bacteria. You only appear to report species that were different in one study rather than a summary of overall findings.

Methods

When was the most recent search carried out? It is not possible to tell if this needs updating based on the information provided.

Lines 103-118 Did they have to report all those measures or just one of them to be included? If the latter please make that clearer.

Lines 120-121 PRISMA guidelines were updated in 2020 (https://www.bmj.com/content/372/bmj.n71) these should be referenced and made sure were adhered to.

Results

Ford et al. 2020 [33] Based on Table 1 it is not clear why this study was included as the association appears to be between diet and microbiome (not sarcopenia). Key findings reported should be relevant to the systematic review question (even if they show null findings).

It would be good to name all microbiota/bacteria that were different in Table 1 rather than generic statements e.g., “A reduced muscle mass was associated with a distinct microbiota” this would help the reader interpret what species, genera etc. may be different.

What confounders were included in each study?

How were the relationships assessed, did they use established cutoffs for low physical function (etc.) compared to normal? This would help the reader interpret the key findings.

Line 184-187 Describe the direction of the associations, were they higher or lower?

Line 208 Rephrase “diseased participants” to be more people-cantered.

Discussion

Only discusses why certain bacteria may be lower but not those that were found to be higher. This should be discussed as well.

Some discussion on why no consistent alteration in bacteria was found should be included.

Confounders in those living with clinical conditions, particularly medications should be discussed.

Given that you encompass so many outcomes would you expect different microbiota signatures depending on the outcome of interest, e.g., low muscle mass vs low grip strength?

Conclusion

The conclusion should make it clear that not all studies found reduction in diversity and/or consistent microbiota that were different in those with low muscle mass etc.

Round 2

Reviewer 1 Report

Comments and Suggestions for Authors

My concerns have not been adequately addressed. As mentioned in my initial review, the critical errors in this study (particularly serious concerns regarding the screening process) cannot be rectified through revisions.

Author Response

Comment:

My concerns have not been adequately addressed. As mentioned in my initial review, the critical errors in this study (particularly serious concerns regarding the screening process) cannot be rectified through revisions.

Response: 

We would like to thank the reviewer again for carefully reading our manuscript. However, we consider the stated limitation that the systematic review does not take original works in all possible languages ​​to be a rather minor problem. The global publication language for original scientific papers is primarily English. Publications in another very frequently used publication language, German, were generally taken into account in the search, but did not yield a single hit. We therefore consider any possible bias caused by the fact that we have reduced the research to a few languages ​​to be very small, if any. We agree with the reviewer that this issue could not be resolved by revising the manuscript and we very much hope that the manuscript can be accepted for publication in this form. However, we added one sentence as a limitation:

"Another possible limitation of the systematic review is that the included original papers were limited to a few publication languages ​​(English and German)."